# Subspace Detours: Building Transport Plans that are Optimal on Subspace Projections

**Boris Muzellec**
CREST, ENSAE
boris.muzellec@ensae.fr

**Marco Cuturi**
Google Brain and CREST, ENSAE
cuturi@google.com

## Abstract

Computing optimal transport (OT) between measures in high dimensions is doomed by the curse of dimensionality. A popular approach to avoid this curse is to project input measures on lower-dimensional subspaces (1D lines in the case of sliced Wasserstein distances), solve the OT problem between these reduced measures, and settle for the Wasserstein distance between these reductions, rather than that between the original measures. This approach is however difficult to extend to the case in which one wants to compute an OT map (a Monge map) between the original measures. Since computations are carried out on lower-dimensional projections, classical map estimation techniques can only produce maps operating in these reduced dimensions. We propose in this work two methods to extrapolate, from an transport map that is optimal on a subspace, one that is nearly optimal in the entire space. We prove that the best optimal transport plan that takes such "subspace detours" is a generalization of the Knothe-Rosenblatt transport. We show that these plans can be explicitly formulated when comparing Gaussian measures (between which the Wasserstein distance is commonly referred to as the Bures or Fréchet distance). We provide an algorithm to select optimal subspaces given pairs of Gaussian measures, and study scenarios in which that mediating subspace can be selected using prior information. We consider applications to semantic mediation between elliptic word embeddings and domain adaptation with Gaussian mixture models.

## 1 Introduction

Minimizing the transport cost between two probability distributions [32] results in two useful quantities: the minimum cost itself, often cast as a loss or a metric (the Wasserstein distance), and the minimizing solution, a function known as the Monge [20] map that pushes forward the first measure onto the second with least expected cost. While the former has long attracted the attention of the machine learning community, the latter is playing an increasingly important role in data sciences. Indeed, important problems such as domain adaptation [8], generative modelling [17, 2, 16], reconstruction of cell trajectories in biology [28] and auto-encoders [19, 30] among others can be recast as the problem of finding a map, preferably optimal, which transforms a reference distribution into another. However, accurately estimating an OT map from data samples is a difficult problem, plagued by the well documented instability of OT in high-dimensional spaces [11, 13] and its high computational cost.

**Optimal Transport on Subspaces.** Several approaches, both in theory and in practice, aim at bridging this gap. Theory [33] supports the idea that sample complexity can be improved when the measures are supported on lower-dimensional manifolds of high-dimensional spaces. Practical insights [9] supported by theory [15] advocate using regularizations to improve both computational and sample complexity. Some regularity in OT maps can also be encoded by looking at specific

families of maps [29, 23]. Another trend relies on lower-dimensional projections of measures before computing OT. In particular, sliced Wasserstein (SW) distances [4] leverage the simplicity of OT between 1D measures to define distances and barycentres, by averaging the optimal transport between projections onto several random directions. This approach has been applied to alleviate training complexity in the GAN/VAE literature [10, 34] and was generalized very recently in [22] who considered projections on $k$-dimensional subspaces that are adversarially selected. However, these subspace approaches only carry out half of the goal of OT: by design, they do result in more robust measures of OT costs, but they can only provide maps in subspaces that are optimal (or nearly so) between the *projected* measures, not transportation maps in the original, high-dimensional space in which the original measures live. For instance, the closest thing to a map one can obtain from using several SW univariate projections is an average of several permutations, which is not a map but a transport plan or coupling [26][25, p.6].

**Our approach.** Whereas the approaches cited above focus on OT maps and plans in projection subspaces only, we consider here plans and maps on the original space that are constrained to be optimal when projected on a given subspace $E$. This results in the definition of a class of transportation plans that figuratively need to make an optimal "detour" in $E$. We propose two constructions to recover such maps corresponding respectively (i) to the independent product between conditioned measures, and (ii) to the optimal conditioned map.

**Paper Structure.** After recalling background material on OT in section 2, we introduce in section 3 the class of *subspace-optimal* plans that satisfy projection constraints on a given subspace $E$. We characterize the degrees of freedom of $E$-optimal plans using their disintegrations on $E$ and introduce two extremal instances: *Monge-Independent* plans, which assume independence of the conditionals, and *Monge-Knothe* maps, in which the conditionals are optimally coupled. We give closed forms for the transport between Gaussian distributions in section 4, respectively as a degenerate Gaussian distribution, and a linear map with block-triangular matrix representation. We provide guidelines and a minimizing algorithm for selecting a subspace $E$ when it is not prescribed *a priori* in section 5. Finally, in section 6 we showcase the behavior of MK and MI transports on (noisy) synthetic data, show how using a mediating subspace can be applied to selecting meanings for polysemous elliptical word embeddings, and experiment using MK maps with the minimizing algorithm on a domain adaptation task with Gaussian mixture models.

**Notations.** For $E$ a linear subspace of $\mathbb{R}^d$, $E^\perp$ is its orthogonal complement, $\mathbf{V}_E \in \mathbb{R}^{d \times k}$ (resp. $\mathbf{V}_{E^\perp} \in \mathbb{R}^{d \times d-k}$) the matrix of orthonormal basis vectors of $E$ (resp $E^\perp$). $p_E : x \to \mathbf{V}_E^\top x$ is the orthogonal projection operator onto $E$. $\mathcal{P}_2(\mathbb{R}^d)$ is the space of probability distributions over $\mathbb{R}^d$ with finite second moments. $\mathcal{B}(\mathbb{R}^d)$ is the Borel algebra over $\mathbb{R}^d$. $\rightharpoonup$ denotes the weak convergence of measures. $\otimes$ is the product of measures, and is used in measure disintegration by abuse of notation.

## 2 Optimal Transport: Plans, Maps and Disintegration of Measure

**Kantorovitch Plans.** For two probability measures $\mu, \nu \in \mathcal{P}_2(\mathbb{R}^d)$, we refer to the set of couplings

$$\Pi(\mu, \nu) \overset{\text{def}}{=} \{\gamma \in \mathcal{P}(\mathbb{R}^d \times \mathbb{R}^d) : \forall A, B \in \mathcal{B}(\mathbb{R}^d), \gamma(A \times \mathbb{R}^d) = \mu(A), \gamma(\mathbb{R}^d \times B) = \nu(B)\}$$

as the set of *transportation plans* between $\mu, \nu$. The 2-Wasserstein distance between $\mu$ and $\nu$ is defined as

$$W_2^2(\mu, \nu) \overset{\text{def}}{=} \min_{\gamma \in \Pi(\mu, \nu)} \mathbb{E}_{(X,Y) \sim \gamma} \left[ \|X - Y\|^2 \right].$$

Conveniently, transportation problems with quadratic cost can be reduced to transportation between centered measures. Indeed, let $\mathbf{m}_\mu$ (resp. $\mathbf{m}_\nu$) denote first moment of $\mu$ (resp. $\nu$). Then, $\forall \gamma \in \Pi(\mu, \nu), \mathbb{E}_{(X,Y) \sim \gamma}[\|X - Y\|^2] = \|\mathbf{m}_\mu - \mathbf{m}_\nu\|^2 + \mathbb{E}_{(X,Y) \sim \gamma}[\|(X - \mathbf{m}_\mu) - (Y - \mathbf{m}_\nu)\|^2]$. Therefore, in the following all probability measures are assumed to be centered, unless stated otherwise.

**Monge Maps.** For a Borel-measurable map $T$, the push-forward of $\mu$ by $T$ is defined as the measure $T_\sharp \mu$ satisfying for all $A \in \mathcal{B}(\mathbb{R}^d), T_\sharp \mu(A) = \mu(T^{-1}(A))$. A map such that $T_\sharp \mu = \nu$ is called a *transportation map* from $\mu$ to $\nu$. When a transportation map exists, the Wasserstein distance can be written in the form of the Monge problem

$$W_2^2(\mu, \nu) = \min_{T: T_\sharp \mu = \nu} \mathbb{E}_{X \sim \mu}[\|X - T(X)\|^2].$$

When it exists, the optimal transportation map $T^\star$ in the Monge problem is called the *Monge map* from $\mu$ to $\nu$. It is then related to the optimal transportation plan $\gamma^\star$ by the relation $\gamma^\star = (\mathrm{Id}, T^\star)_\sharp \mu$. When $\mu$ and $\nu$ are absolutely continuous (a.c.), a Monge map always exists ([27], Theorem 1.22).

**Global Maps or Plans that are Locally Optimal.** Considering the projection operator on $E$, $p_E$, we write $\mu_E = (p_E)_\sharp \mu$ for the marginal distribution of $\mu$ on $E$. Suppose that we are given a Monge map $S$ between the two projected measures $\mu_E$ and $\nu_E$. One of the contributions of this paper is to propose extensions of this map $S$ as a transportation plan $\gamma$ (resp. a new map $T$) whose projection $\gamma_E = (p_E, p_E)_\sharp \gamma$ on that subspace $E$ coincides with the optimal transportation plan $(\mathrm{Id}_E, S)_\sharp \mu_E$ (resp. $p_E \circ T = S \circ p_E$). Formally, the transports introduced in section 3 only require that $S$ be a transport map from $\mu_E$ to $\nu_E$, but optimality is required in the closed forms given in section 4 for Gaussian distributions. In either case, this constraint implies that $\gamma$ is built "assuming that" it is equal to $(\mathrm{Id}_E, S)_\sharp \mu_E$ on $E$. This is rigorously defined using the notion of measure disintegration.

**Disintegration of Measures.** The disintegration of $\mu$ on a subspace $E$ is the collection of measures $(\mu_{x_E})_{x_E \in E}$ supported on the fibers $\{x_E\} \times E^\perp$ such that any test function $\phi$ can be integrated against $\mu$ as $\int_{\mathbb{R}^d} \phi \mathrm{d}\mu = \int_E \left( \int_{E^\perp} \phi(y) \mathrm{d}\mu_{x_E}(y) \right) \mathrm{d}\mu_E(x_E)$. In particular, if $X \sim \mu$, then the law of $X$ given $x_E$ is $\mu_{x_E}$. By abuse of the measure product notation $\otimes$, measure disintegration is denoted as $\mu = \mu_{x_E} \otimes \mu_E$. A more general description of disintegration can be found in [1], Ch. 5.5.

# 3 Lifting Transport from Subspace to Full Space

Given two distributions $\mu, \nu \in \mathcal{P}_2(\mathbb{R}^d)$, it is often easier to compute a Monge map $S$ between their marginals $\mu_E, \nu_E$ on a $k$-dimensional subspace $E$ rather than in the whole space $\mathbb{R}^d$. When $k = 1$, this fact is at the heart of sliced wasserstein approaches [4], which have recently sparked interest in the GAN/VAE literature [10, 34]. However, when $k < d$, there is in general no straightforward way of extending $S$ to a transportation map or plan between $\mu$ and $\nu$. In this section, we prove the existence of such extensions and characterize them.

**Subspace-Optimal Plans.** A transportation plan between $\mu_E$ and $\nu_E$ is a coupling living in $\mathcal{P}(E \times E)$. In general, it cannot be cast directly as a transportation plan between $\mu$ and $\nu$ taking values in $\mathcal{P}(\mathbb{R}^d \times \mathbb{R}^d)$. However, the existence of such a "lifted" plan is given by the following result, which is used in OT theory to prove that $W_p$ is a metric:

**Lemma 1** (The Gluing Lemma, [32]). *Let $\mu_1, \mu_2, \mu_3 \in \mathcal{P}(\mathbb{R}^d)$. If $\gamma_{12}$ is a coupling of $(\mu_1, \mu_2)$ and $\gamma_{23}$ is a coupling of $(\mu_2, \mu_3)$, then one can construct a triple of random variables $(Z_1, Z_2, Z_3)$ such that $(Z_1, Z_2) \sim \gamma_{12}$ and $(Z_2, Z_3) \sim \gamma_{23}$.*

By extension of the lemma, if we define (i) a coupling between $\mu$ and $\mu_E$, (ii) a coupling between $\nu$ and $\nu_E$, and (iii) the optimal coupling between $\mu_E$ and $\nu_E$, $(\mathrm{Id}, S)_\sharp \mu_E$ (where $S$ stands for the Monge map from $\mu_E$ to $\nu_E$), we get the existence of four random variables (with laws $\mu, \mu_E, \nu$ and $\nu_E$) which follow the desired joint laws. However, the lemma does not imply the uniqueness of those random variables, nor does it give a closed form for the corresponding coupling between $\mu$ and $\nu$.

**Definition 1** (Subspace-Optimal Plans). *Let $\mu, \nu \in \mathcal{P}_2(\mathbb{R}^d)$ and $E$ be a $k$-dimensional subspace of $\mathbb{R}^d$. Let $S$ be a Monge map from $\mu_E$ to $\nu_E$. We define the set of $E$-optimal plans between $\mu$ and $\nu$ as $\Pi_E(\mu, \nu) \overset{\mathrm{def}}{=} \{\gamma \in \Pi(\mu, \nu) : \gamma_E = (\mathrm{Id}_E, S)_\sharp \mu_E\}$.*

**Degrees of freedom in $\Pi_E(\mu, \nu)$.** When $k < d$, there can be infinitely many $E$-optimal plans. However, we can further characterize the degrees of freedom available to define plans in $\Pi_E(\mu, \nu)$. Indeed, let $\gamma \in \Pi_E(\mu, \nu)$. Then, disintegrating $\gamma$ on $E \times E$, we get $\gamma = \gamma_{(x_E, y_E)} \otimes \gamma_E$, i.e. plans in $\Pi_E(\mu, \nu)$ only differ on their disintegrations on $E \times E$. Further, since $\gamma_E$ stems from a transport (Monge) map $S$, it is supported on the graph of $S$ on $E$, $\mathcal{G}(S) = \{(x_E, S(x_E)) : x_E \in E\} \subset E \times E$. This implies that $\gamma$ puts zero mass when $y_E \neq S(x_E)$ and thus that $\gamma$ is fully characterized by $\gamma_{(x_E, S(x_E))}, x_E \in E$, i.e. by the couplings between $\mu_{x_E}$ and $\nu_{S(x_E)}$ for $x_E \in E$. This is illustrated in Figure 1. Two such couplings are presented: the first, MI (Definition 2) corresponds to independent couplings between the conditionals, while the second (MK, Definition 3) corresponds to optimal couplings between the conditionals.

**Definition 2** (Monge-Independent Plans). $\pi_{MI} \overset{\mathrm{def}}{=} (\mu_{x_E} \otimes \nu_{S(x_E)}) \otimes (\mathrm{Id}_E, S)_\sharp \mu_E$.

Monge-Independent transport only requires that there exists a Monge map $S$ between $\mu_E$ and $\nu_E$ (and not on the whole space), but extends $S$ as a transportation plan and not a map. Since it couples disintegrations with the independent law, it is particularly suited to settings where all the information is contained in $E$, as shown in section 6.

When there exists a Monge map between disintegrations $\mu_{x_E}$ to $\nu_{S(x_E)}$ for all $x_E \in E$ (e.g. when $\mu$ and $\nu$ are a.c.), it is possible to extend $S$ as a transportation map between $\mu$ and $\nu$ using those maps. Indeed, for all $x_E \in E$, let $\hat{T}(x_E; \cdot) : E^\perp \to E^\perp$

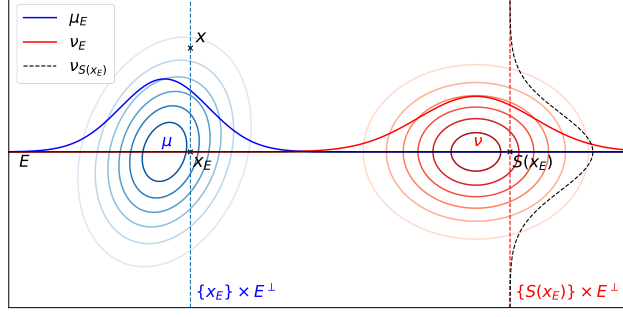

Figure 1: A $d = 2, k = 1$ illustration. Any $\gamma \in \Pi_E(\mu, \nu)$ being supported on $\mathcal{G}(S) \times (E^\perp)^2$, all the mass from $x$ is transported on the fiber $\{S(x_E)\} \times E^\perp$. Different $\gamma$'s in $\Pi_E(\mu, \nu)$ correspond to different couplings between the fibers $\{x_E\} \times E^\perp$ and $\{S(x_E)\} \times E^\perp$.

denote the Monge map from $\mu_{x_E}$ to $\nu_{S(x_E)}$. The *Monge-Knothe* transport corresponds to the $E$-optimal plan with optimal couplings between the disintegrations:

**Definition 3** (Monge-Knothe Transport). $T_{MK}(x_E, x_{E^\perp}) \stackrel{\text{def}}{=} (S(x_E), \hat{T}(x_E; x_{E^\perp})) \in E \oplus E^\perp$.

The proof that $T_{\text{MK}}$ defines a transport map from $\mu$ to $\nu$ is a direct adaptation of the proof for the Knothe-Rosenblatt transport ([27], Section 2.3). When it is not possible to define a Monge map between the disintegrations, one can still consider the optimal couplings $\pi_{\text{OT}}(\mu_{x_E}, \nu_{S(x_E)})$ and define $\pi_{\text{MK}} = \pi_{\text{OT}}(\mu_{x_E}, \nu_{S(x_E)}) \otimes (\text{Id}_E, S)_\sharp \mu_E$, which we still call Monge-Knothe plan by abuse. In either case, $\pi_{\text{MK}}$ is the $E$-optimal plan with lowest global cost:

**Proposition 1.** *The Monge-Knothe plan is optimal in $\Pi_E(\mu, \nu)$, namely*

$$\pi_{MK} \in \underset{\gamma \in \Pi_E(\mu, \nu)}{\arg\min} \mathbb{E}_{(X,Y) \sim \gamma}[\|X - Y\|^2].$$

*Proof.* $E$-optimal plans only differ in the couplings they induce between $\mu_{x_E}$ and $\nu_{S(x_E)}$ for $x_E \in E$. Since $\pi_{\text{MK}}$ corresponds to the case when these couplings are optimal, disintegrating $\gamma$ over $E \times E$ in $\int_{\mathbb{R}^d \times \mathbb{R}^d} \|x - y\|^2 d\gamma(x, y)$ shows that $\gamma = \pi_{\text{MK}}$ has the lowest cost. ∎

**Relation with the Knothe-Rosenblatt (KR) transport.** These definitions are related to the KR transport ([27], section 2.3), which consists in defining a transport map between two a.c. measures by recursively (i) computing the Monge map $T_1$ between the first two one-dimensional marginals of $\mu$ and $\nu$ and (ii) repeating the process between the disintegrated measures $\mu_{x_1}$ and $\nu_{T_1(x_1)}$. MI and MK marginalize on the $k \geq 1$ dimensional subspace $E$, and respectively define the transport between disintegrations $\mu_{x_E}$ and $\nu_{S(x_E)}$ as the product measure and the optimal transport instead of recursing.

**MK as a limit of optimal transport with re-weighted quadratic costs.** Similarly to KR [5], MK transport maps can intuitively be obtained as the limit of optimal transport maps, when the costs on $E^\perp$ become negligible compared to the costs on $E$.

**Proposition 2.** *Let $\mathbb{R}^d = E \oplus E^\perp$, $(\mathbf{V}_E \quad \mathbf{V}_{E^\perp})$ an orthonormal basis of $E \oplus E^\perp$ and $\mu, \nu \in \mathcal{P}_2(\mathbb{R}^d)$ be two a.c. probability measures. Define*

$$\forall \varepsilon > 0, \quad \mathbf{P}_\varepsilon \stackrel{\text{def}}{=} \mathbf{V}_E \mathbf{V}_E^\top + \varepsilon \mathbf{V}_{E^\perp} \mathbf{V}_{E^\perp}^\top, \quad d^2_{\mathbf{P}_\varepsilon}(x, y) \stackrel{\text{def}}{=} (x - y)^\top \mathbf{P}_\varepsilon (x - y).$$

*Let $T_\varepsilon$ be the optimal transport map for the cost $d^2_{\mathbf{P}_\varepsilon}$. Then $T_\varepsilon \to T_{MK}$ in $\mathrm{L}_2(\mu)$.*

Proof in the supplementary material.

**MI as a limit of the discrete case.** When $\mu$ and $\nu$ are a.c., for $n \in \mathbb{N}$ let $\mu_n, \nu_n$ denote the uniform distribution over $n$ i.i.d. samples from $\mu$ and $\nu$ respectively, and let $\pi_n$ be an optimal transportation plan between $(p_E)_\sharp \mu_n$ and $(p_E)_\sharp \nu_n$ given by a Monge map (which is possible assuming uniform weights and non-overlapping projections). We have that $\mu_n \rightharpoonup \mu$ and $\nu_n \rightharpoonup \mu$. From [27], Th 1.50, 1.51, we have that $\pi_n \in \mathcal{P}_2(E \times E)$ converges weakly, up to subsequences, to a coupling $\pi \in \mathcal{P}_2(E \times E)$ that is optimal for $\mu_E$ and $\nu_E$. On the other hand, up to points having the same

projections, the discrete plans $\pi_n$ can also be seen as plans in $\mathcal{P}(\mathbb{R}^d \times \mathbb{R}^d)$. A natural question is then whether the sequence $\pi_n \in \mathcal{P}(\mathbb{R}^d \times \mathbb{R}^d)$ has a limit in $\mathcal{P}(\mathbb{R}^d \times \mathbb{R}^d)$.

**Proposition 3.** *Let $\mu, \nu \in \mathcal{P}_2(\mathbb{R}^d)$ be a.c. and compactly supported, $\mu_n, \nu_n, n \geq 0$ be uniform distributions over $n$ i.i.d. samples, and $\pi_n \in \Pi_E(\mu_n, \nu_n), n \geq 0$. Then $\pi_n \rightharpoonup \pi_{MI}(\mu, \nu)$.*

Proof in the supplementary material. We conjecture that under additional assumptions, the compactness hypothesis can be relaxed. In particular, we empirically observe convergence for Gaussians.

# 4 Explicit Formulas for Subspace Detours in the Bures Metric

Multivariate Gaussian measures are a specific case of continuous distributions for which Wasserstein distances and Monge maps are available in closed form. We first recall basic facts about optimal transport between Gaussian measures, and then show that the $E$-optimal transports MI and MK introduced in section 3 are also in closed form. For two Gaussians $\mu, \nu$, one has $W_2^2(\mu, \nu) = \|\mathbf{m}_\mu - \mathbf{m}_\nu\|^2 + \mathfrak{B}^2(\text{var } \mu, \text{var } \nu)$ where $\mathfrak{B}$ is the *Bures* metric [3] between PSD matrices [14]: $\mathfrak{B}^2(\mathbf{A}, \mathbf{B}) \overset{\text{def}}{=} \text{Tr} \mathbf{A} + \text{Tr} \mathbf{B} - 2\text{Tr}(\mathbf{A}^{1/2}\mathbf{B}\mathbf{A}^{1/2})^{1/2}$. The Monge map from a centered Gaussian distribution $\mu$ with covariance matrix $\mathbf{A}$ to one $\nu$ with covariance matrix $\mathbf{B}$ is linear and is represented by the matrix $\mathbf{T}^{\mathbf{AB}} \overset{\text{def}}{=} \mathbf{A}^{-1/2}(\mathbf{A}^{1/2}\mathbf{B}\mathbf{A}^{1/2})^{1/2}\mathbf{A}^{-1/2}$. For any linear transport map, $\mathbf{T}_\sharp \mu$ has covariance $\mathbf{T}\mathbf{A}\mathbf{T}^\top$, and the transportation cost from $\mu$ to $\nu$ is $\mathbb{E}_{X \sim \mu}[\|X - \mathbf{T}X\|^2] = \text{Tr} \mathbf{A} + \text{Tr} \mathbf{B} - \text{Tr}(\mathbf{TA} + \mathbf{AT}^\top)$. In the following, $\mu$ (resp. $\nu$) will denote the centered Gaussian distribution with covariance matrix $\mathbf{A}$ (resp. $\mathbf{B}$). We write $\mathbf{A} = \begin{pmatrix} \mathbf{A}_E & \mathbf{A}_{EE^\perp} \\ \mathbf{A}_{EE^\perp}^\top & \mathbf{A}_{E^\perp} \end{pmatrix}$ when $\mathbf{A}$ is represented in an orthonormal basis $(\mathbf{V}_E \quad \mathbf{V}_{E^\perp})$ of $E \oplus E^\perp$.

**Monge-Independent Transport for Gaussians.** The MI transport between Gaussian measures is given by a degenerate Gaussian, i.e. a measure with Gaussian density over the image of its covariance matrix $\Sigma$ (we refer to the supplementary material for the proof).

**Proposition 4** (Monge-Independent (MI) Transport for Gaussians)**.** *Let*

$$\mathbf{C} \overset{\text{def}}{=} \left( \mathbf{V}_E \mathbf{A}_E + \mathbf{V}_{E^\perp} \mathbf{A}_{EE^\perp}^\top \right) \mathbf{T}^{\mathbf{A}_E \mathbf{B}_E} \left( \mathbf{V}_{E^\top} + (\mathbf{B}_E)^{-1} \mathbf{B}_{EE^\perp} \mathbf{V}_{E^\perp}^\top \right) \text{ and } \quad \Sigma \overset{\text{def}}{=} \begin{pmatrix} \mathbf{A} & \mathbf{C} \\ \mathbf{C}^\top & \mathbf{B} \end{pmatrix}.$$

*Then $\pi_{MI}(\mu, \nu) = \mathcal{N}(0_{2d}, \Sigma) \in \mathcal{P}(\mathbb{R}^d \times \mathbb{R}^d)$.*

**Knothe-Rosenblatt and Monge-Knothe for Gaussians.** Before giving the closed-form MK map for Gaussian measures, we derive the KR map ([27], section 2.3) with successive marginalization[1] on $x_1, x_2, ..., x_d$. When $d = 2$ and the basis is orthonormal for $E \oplus E^\perp$, those two notions coincide.

**Proposition 5** (Knothe-Rosenblatt (KR) Transport between Gaussians)**.** *Let $\mathbf{L}_A$ (resp. $\mathbf{L}_B$) be the Cholesky factor of $\mathbf{A}$ (resp. $\mathbf{B}$). The KR transport from $\mu$ to $\nu$ is a linear map whose matrix is given by $\mathbf{T}_{KR}^{\mathbf{AB}} = \mathbf{L}_B(\mathbf{L}_A)^{-1}$. Its cost is the squared Frobenius distance between the Cholesky factors $\mathbf{L}_A$ and $\mathbf{L}_B$:*

$$\mathbb{E}_{X \sim \mu}[\|X - T_{KR}^{\mathbf{AB}}X\|^2] = \|\mathbf{L}_A - \mathbf{L}_B\|^2.$$

*Proof.* The KR transport with successive marginalization on $x_1, x_2, ..., x_d$ between two a.c. distributions has a lower triangular Jacobian with positive entries on the diagonal. Further, since the one-dimensional disintegrations of Gaussians are Gaussians themselves, and since Monge maps between Gaussians

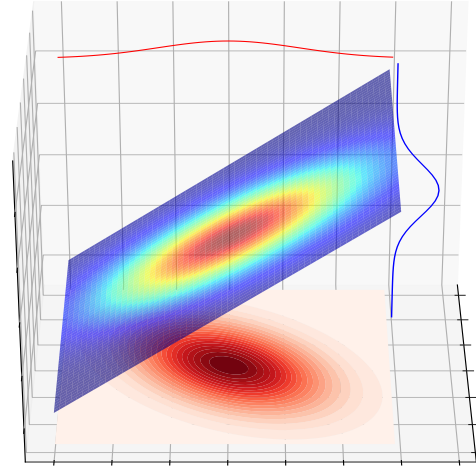

Figure 2: MI transport from a 2D Gaussian (red) to a 1D Gaussian (blue), projected on the $x$-axis. The two 1D distributions represent the projections of both Gaussians on the $x$-axis, the blue one being already originally supported on the $x$-axis. The oblique hyperplane is the support of $\pi_{\text{MI}}$, onto which its density is represented.

are linear, the KR transport between two centered Gaussians is a linear map, hence its matrix representation equals its Jacobian and is lower triangular.

Let $\mathbf{T} = \mathbf{L}_B(\mathbf{L}_A)^{-1}$. We have $\mathbf{T}\mathbf{A}\mathbf{T}^\top = \mathbf{L}_B\mathbf{L}_A^{-1}\mathbf{L}_A\mathbf{L}_A^\top\mathbf{L}_A^{-\top}\mathbf{L}_B^\top = \mathbf{L}_B\mathbf{L}_B^\top = \mathbf{B}$, i.e. $\mathbf{T}_\sharp\mu = \nu$. Further, since $\mathbf{T}\mathbf{L}_A$ is the Cholesky factor for $\mathbf{B}$, and since $\mathbf{A}$ is supposed non-singular, by unicity of the Cholesky decomposition $\mathbf{T}$ is the only lower triangular matrix satisfying $\mathbf{T}_\sharp\mu = \nu$. Hence, it is the KR transport map from $\mu$ to $\nu$.

Finally, we have that $\mathbb{E}_{X\sim\mu}[\|X - \mathbf{T}_{\mathrm{KR}}X\|^2] = \mathrm{Tr}(\mathbf{A} + \mathbf{B} - (\mathbf{A}(\mathbf{T}_{\mathrm{KR}})^\top + \mathbf{T}_{\mathrm{KR}}\mathbf{A})) = \mathrm{Tr}(\mathbf{L}_A\mathbf{L}_A^\top + \mathbf{L}_B\mathbf{L}_B^\top - (\mathbf{L}_A\mathbf{L}_B^\top + \mathbf{L}_B\mathbf{L}_A^\top)) = \|\mathbf{L}_A - \mathbf{L}_B\|^2$ ∎

**Corollary 1.** *The (square root) cost of the Knothe-Rosenblatt transport $(\mathbb{E}_{X\sim\mu}[\|X - \mathbf{T}_{KR}X\|^2])^{1/2}$ between centered gaussians defines a distance (i.e. it satisfies all three metric axioms).*

*Proof.* This comes from the fact that $\left(\mathbb{E}_{X\sim\mu}[\|X - \mathbf{T}_{\mathrm{KR}}X\|^2]\right)^{1/2} = \|\mathbf{L}_A - \mathbf{L}_B\|$. ∎

As can be expected from the fact that MK can be seen as a generalization of KR, the MK transportation map is linear and has a block-triangular structure. The next proposition shows that the MK transport map can be expressed as a function of the Schur complements $\mathbf{A}/\mathbf{A}_E \stackrel{\text{def}}{=} \mathbf{A}_{E^\perp} - \mathbf{A}_{EE^\perp}^\top\mathbf{A}_E^{-1}\mathbf{A}_{EE^\perp}$ of $\mathbf{A}$ w.r.t. $\mathbf{A}_E$, and $\mathbf{B}$ w.r.t. $\mathbf{B}_E$, which are the covariance matrices of $\mu$ (resp. $\nu$) conditioned on $E$.

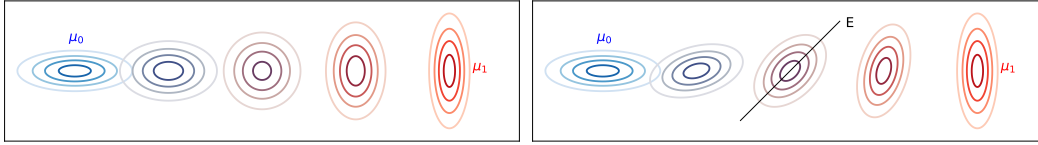

(a) Usual Monge Interpolation of Gaussians      (b) Monge-Knothe Interpolation through E

Figure 3: (a) Wasserstein-Bures geodesic and (b) Monge-Knothe interpolation through $E = \{(x,y) : x = y\}$ from $\mu_0$ to $\mu_1$, at times $t = 0, 0.25, 0.5, 0.75, 1$.

**Proposition 6** (Monge-Knothe (MK) Transport for Gaussians)**.** *Let $\mathbf{A}, \mathbf{B}$ be represented in an orthonormal basis for $E \oplus E^\perp$. The MK transport map on $E$ between $\mu$ and $\nu$ is given by*

$$\mathbf{T}_{\mathrm{MK}} = \begin{pmatrix} \mathbf{T}^{\mathbf{A}_E\mathbf{B}_E} & 0_{k\times(d-k)} \\ \left[\mathbf{B}_{EE^\perp}^\top(\mathbf{T}^{\mathbf{A}_E\mathbf{B}_E})^{-1} - \mathbf{T}^{(\mathbf{A}/\mathbf{A}_E)(\mathbf{B}/\mathbf{B}_E)}\mathbf{A}_{EE^\perp}^\top\right](\mathbf{A}_E)^{-1} & \mathbf{T}^{(\mathbf{A}/\mathbf{A}_E)(\mathbf{B}/\mathbf{B}_E)} \end{pmatrix}.$$

*Proof.* As can be seen from the structure of the MK transport map in Definition 3, $T_{\mathrm{MK}}$ has a lower block-triangular Jacobian (with block sizes $k$ and $d - k$), with PSD matrices on the diagonal (corresponding to the Jacobians of the Monge maps (i) between marginals and (ii) between conditionals). Further, since $\mu$ and $\nu$ are Gaussian, their disintegrations are Gaussian as well. Hence, all Monge maps from the disintegrations of $\mu$ to that of $\nu$ are linear, and therefore the matrix representing $\mathbf{T}$ is equal to its Jacobian. One can check that the map $\mathbf{T}$ in the proposition verifies $\mathbf{T}\mathbf{A}\mathbf{T}^\top = \mathbf{B}$ and is of the right form. One can also check that it is the unique such matrix, hence it is the MK transport map. ∎

## 5   Selecting the Supporting Subspace

Both MI and MK transports are highly dependent on the chosen subspace $E$. Depending on applications, $E$ can either be prescribed (e.g. if one has access to a transport map between the marginals in a given subspace) or has to be selected. In the latter case, we give guidelines on how prior knowledge can be used, and alternatively propose an algorithm for minimizing the MK distance.

**Subspace Selection Using Prior Knowledge.**   When prior knowledge is available, one can choose a mediating subspace $E$ to enforce specific criteria when comparing two distributions. Indeed, if the directions in $E$ are known to correspond to given properties of the data, then MK or MI transport privileges those properties when matching distributions over those not encoded by $E$. In particular, if one has access to features $\mathbf{X}$ from a reference dataset, one can use

---

**Algorithm 1** MK Subspace Selection

**Input:** $\mathbf{A}, \mathbf{B} \in \mathrm{PSD}, k \in [\![1, d]\!], \eta$
  $\mathbf{V} \leftarrow \mathrm{Polar}(\mathbf{A}\mathbf{B})$
  **while** not converged **do**
    $\mathcal{L} \leftarrow \mathrm{MK}(\mathbf{V}^\top\mathbf{A}\mathbf{V}, \mathbf{V}^\top\mathbf{B}\mathbf{V}; k)$
    $\mathbf{V} \leftarrow \mathbf{V} - \eta\nabla_{\mathbf{V}}\mathcal{L}$
    $\mathbf{V} \leftarrow \mathrm{Polar}(\mathbf{V})$
  **end while**
**Output:** $E = \mathrm{Span}\{\mathbf{v}_1, .., \mathbf{v}_k\}$

---

principal component analysis (PCA) and select the first $k$ principal directions to compare datasets $\mathbf{X}_1$ and $\mathbf{X}_2$. MK and MI then allow comparing $\mathbf{X}_1$ and $\mathbf{X}_2$ using the most significant features from the reference $\mathbf{X}$ with higher priority. In section 6, we experiment this method on word embeddings.

**Minimal Monge-Knothe Subspace.** Alternatively, in the absence of prior knowledge, it is natural to aim at finding the subspace which minimizes MK. Unfortunately, optimization on the Grassmann manifold is quite hard in general, which makes direct optimization of MK w.r.t. $E$ impractical. Optimizing with respect to an orthonormal matrix $\mathbf{V}$ of basis vectors of $\mathbb{R}^d$ is a more practical parameterization, which allows to perform projected gradient descent (Algorithm 1). The projection step consists in computing a polar decomposition, as the projection of a matrix $\mathbf{V}$ onto the set of unitary matrices is the unitary matrix in the polar decomposition of $\mathbf{V}$. The proposed initialization is $V = \text{Polar}(\mathbf{AB})$, as this is the optimal solution when $\mathbf{A}, \mathbf{B}$ are co-diagonalizable. Note that since the function being minimized is non-convex, Algorithm 1 is only guaranteed to converge to a local minimum. In section 6, experimental evaluation of Algorithm 1 is carried out on noise-contaminated synthetic data (Figure 6) and on a domain adaptation task with Gaussian mixture models on the Office Home dataset [31] with inception features (Figure 7).

# 6  Experiments

**Color Transfer.** Given a source and a target image, the goal of color transfer is to map the color palette of the source image (represented by its RGB histogram) into that of the target image. A natural toolbox for such a task is optimal transport, see *e.g.* [4, 12, 24]. First, a

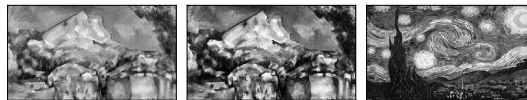

(a) Gray Source    (b) Gray OT    (c) Gray Target

Figure 4: OT color transfer between gray projections.

k-means quantization of both images is computed. Then, the colors of the pixels within each source cluster are modified according to the optimal transport map between both color distributions. In Figure 5, we illustrate discrete MK transport maps for color transfer. In this setting, we project images on the 1D space of grayscale images, relying on the 1D OT sorting-based algorithm (Figure 4). Then, we solve small 2D OT problems on the corresponding disintegrations. We compare this approach with classic full OT maps and a sliced OT approach (with 100 random projections). As can be seen in Figure 5, MK results are visually very similar to that of full OT, with a x50 speedup allowed by the fast 1D OT sorting-based algorithm that is comparable to sliced OT.

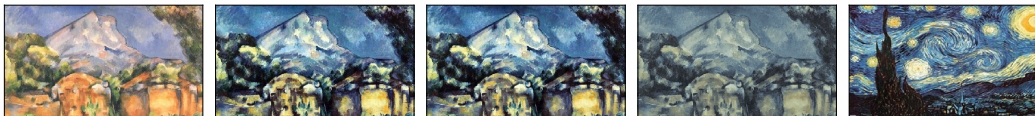

(a) Source    (b) Full OT (2.67s)    (c) Gray MK (0.052s)    (d) Sliced (0.057s)    (e) Target

Figure 5: Color transfer, after quantization using 3000 k-means clusters, with corresponding runtimes.

**Synthetic Data.** We test the behavior of MK and MI in a noisy environment, where the signal is supported in a subspace of small dimension. We represent the signal using two normalized PSD matrices $\mathbf{A}, \mathbf{B} \in \mathbb{R}^{d_1 \times d_1}$ and sample noise $\Sigma_1, \Sigma_2 \in \mathbb{R}^{d_2 \times d_2}, d_2 \geq d_1$ from a Wishart distribution with parameter $\mathbf{I}$. We then build the noisy covariance $\mathbf{A}_\varepsilon = \left( \begin{smallmatrix} \mathbf{A} & 0 \\ 0 & 0 \end{smallmatrix} \right) + \varepsilon \Sigma_1 \in \mathbb{R}^{d_2 \times d_2}$ (and likewise $\mathbf{B}_\varepsilon$) for different noise levels $\varepsilon$ and compute MI and MK distances along the first $k$ directions, $k = 1, ..., d_2$. As can be seen in Figure 6, both MI and MK curves exhibit a local minimum or an "elbow" when $k = d_1$, i.e. when $E$ corresponds to the subspace where the signal is located. However, important differences in the behaviors of MI and MK can be noticed. Indeed, MI has a steep decreasing curve from 1 to $d_1$ and then a slower decreasing curve. This is explained by the fact that MI transport computes the OT map along the $k$ directions of $E$ only, and treats the conditionals as being independent. Therefore, if $k \geq d_1$, all the signal has been fitted and for increasing values of $k$ MI starts fitting the noise as well. On the other hand, MK transport computes the optimal transport on both $E$ and the corresponding $(d_2 - k)$-dimensional conditionals. Therefore, if $k \neq d_1$, either or both maps fit a mixture of signal and noise. Local maxima correspond to cases where the signal is the most contaminated by noise, and minima $k = d_1$, $k = d_2$ to cases where either the marginals or the conditionals are unaffected by noise. Using Algorithm 1 instead of the principle directions allows to find better subspaces than the first $k$ directions when $k \leq d_1$, and then behaves similarly (up to the gradient being stuck in local minima and thus being occasionally less competitive). Overall, the

differences in behavior of MI and MK show that MI is more adapted to noisy environments, and MK to applications where all directions are meaningful, but where one wishes to prioritize fitting on a subset of those directions, as shown in the next experiment.

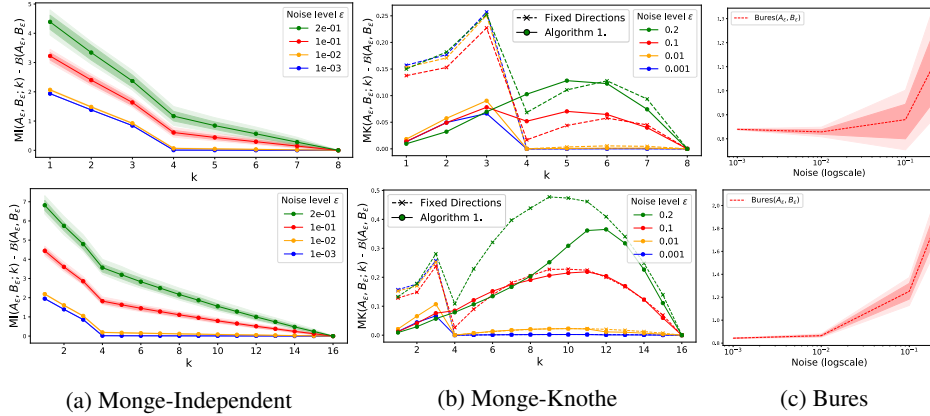

|       (a) Monge-Independent       |       (b) Monge-Knothe       |       (c) Bures       |

Figure 6: (a)-(b): Difference between (a) MI and Bures and (b) MK and Bures metrics for different noise levels $\varepsilon$ and subspace dimensions $k$. (c): Corresponding Bures values. For each $\epsilon$, 100 different noise matrices are sampled. Points show mean values, and shaded areas the 25%-75% and 10%-90% percentiles. Top row: $d_1 = 4, d_2 = 8$. Bottom row: $d_1 = 4, d_2 = 16$.

**Semantic Mediation.** We experiment using reference features for comparing distributions with elliptical word embeddings [21], which represent each word from a given corpus using a mean vector and a covariance matrix. For a given embedding, we expect the principal directions of its covariance matrix to be linked to its semantic content. Therefore, the comparison of two words $w_1, w_2$ based on the principal eigenvectors of a context word $c$ should be impacted by the semantic relations of $w_1$ and $w_2$ with respect to $c$, e.g. if $w_1$ is polysemous and $c$ is related to a specific meaning. To test this intuition, we compute the nearest neighbors of a given word $w$ according to the MK distance with $E$ taken as the subspace spanned by the principal directions of two different contexts $c_1$ and $c_2$. We exclude means and compute MK based on covariances only, and look at the symmetric difference of the returned sets of words (i.e. words in $\text{KNN}(w|c_1)$ but not in $\text{KNN}(w|c_2)$, and inversely). Table 1 shows that specific contexts affect the nearest neighbors of ambiguous words.

Table 1: Symmetric differences of the 20-NN sets of $w$ given $c_1$ minus $w$ given $c_2$ using MK. Embeddings are $12 \times 12$ pretrained normalized covariance matrices from [21]. $E$ is spanned by the 4 principal directions of the contexts. Words are printed in increasing distance order.

| Word | Context 1 | Context 2 | Difference |
|------|-----------|-----------|------------|
| instrument | monitor | oboe | cathode, monitor, sampler, rca, watts, instrumentation, telescope, synthesizer, ambient |
|  | oboe | monitor | tuned, trombone, guitar, harmonic, octave, baritone, clarinet, saxophone, virtuoso |
| windows | pc | door | netscape, installer, doubleclick, burner, installs, adapter, router, cpus |
|  | door | pc | screwed, recessed, rails, ceilings, tiling, upvc, profiled, roofs |
| fox | media | hedgehog | Penny, quiz, Whitman, outraged, Tinker, ads, Keating, Palin, show |
|  | hedgehog | media | panther, reintroduced, kangaroo, Harriet, fair, hedgehog, bush, paw, bunny |

**MK Domain Adaptation with Gaussian Mixture Models.** Given a *source* dataset of labeled data, domain adaptation (DA) aims at finding labels for a *target* dataset by transfering knowledge from the source. Such a problem has been successfully tackled using OT-based techniques [8]. We illustrate using MK Gaussian maps on a domain adaptation task where both source and target distributions are modeled by a Gaussian mixture model (GMM). We use the Office Home dataset [31], which comprises 15000 images from 65 different classes across 4 domains: `Art`, `Clipart`, `Product` and `Real World`. For each image, we consider 2048-dimensional features taken from the coding layer of an inception model, as with Fréchet inception distances [18]. For each source/target pair, we represent the source as a GMM by fitting one Gaussian per source class and defining mixture weights proportional to class frequencies, and we fit a GMM with the same number of components on the target. Since label information is not available for the target dataset, data from different classes may be assigned to the same component. We then compute pairwise MK distances between all source and target components, and solve for the discrete OT plan $P$ using those distances as costs and mixture weights as marginals (as in [6] with Bures distances). Finally, we map the source distribution on the target by computing the $P$-barycentric projection of the component-wise MK maps $\sum_{ij} P_{ij} T_{\text{MK}}^{ij}$,

and assign target labels using 1-NN prediction over the mapped source data. The same procedure is applied using Bures distances between the projections on $E$. We use Algorithm 1 between the empirical covariance matrices of the source and target datasets to select the supporting subspace $E$, for different values of the supporting dimension $k$ (Figure 7).

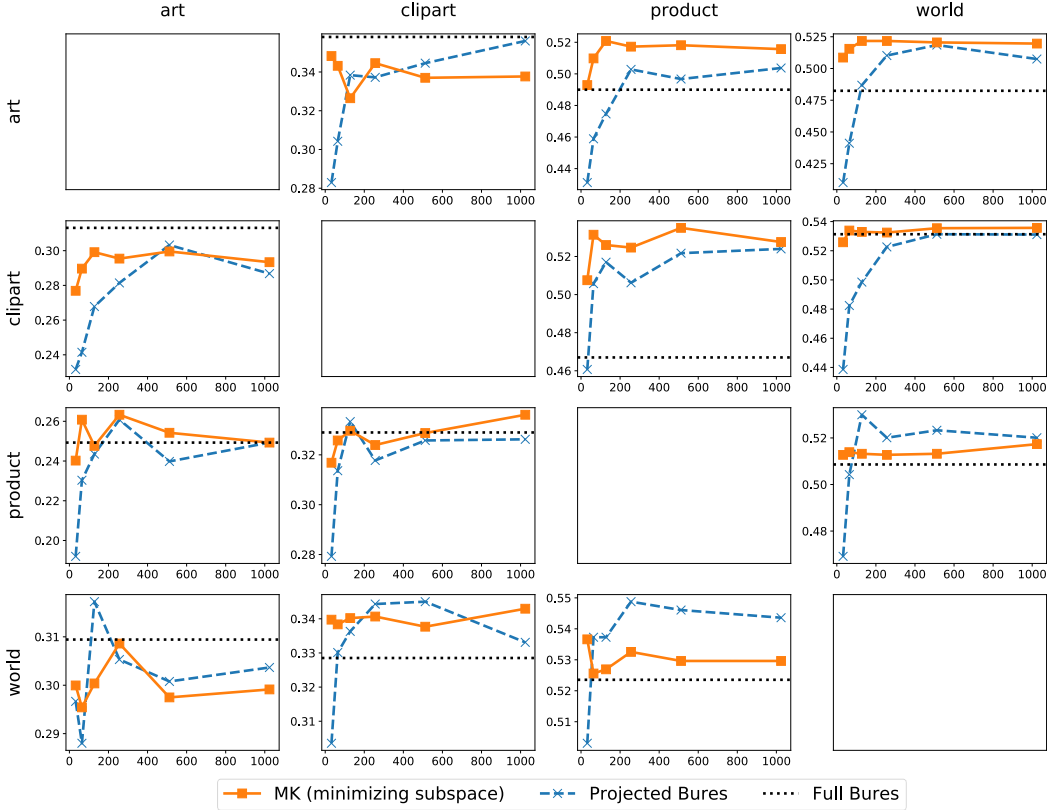

Figure 7: Domain Adaptation: 1-NN accuracy scores on the Office Home dataset v.s. dimension $k$. We compare the $k$-dimensional projected Bures maps with the $E$-MK maps and the 2048-D Bures baseline. $E$ is selected using Algorithm 1 between the source and target covariance matrices for $k = 32, 64, 128, 256, 512, 1024$. Rows: sources, Columns: targets.

Several facts can be observed from Figure 7. First, using the full 2048-dimensional Bures maps is regularly sub-optimal compared to Bures (resp. MK) maps on a lower-dimensional subspace, even though this is dependent on the source/target combination. This shows the interest of not using all available features equally in transport problems. Secondly, when $E$ is chosen using the minimizing algorithm 1, in most cases MK maps yield equivalent or better classification accuracy that the corresponding Bures maps on the projections, even though they have the same projections on $E$. However, as can be expected, this does not hold for an arbitrary choice of $E$ (not shown in the figure). Due to the relative simplicity of this DA method (which models the domains as GMMs), we do not aim at comparing with state-of-the-art OT DA methods [8, 7] (which compute transportation plans between the discrete distributions directly). The goal is rather to illustrate how MK maps can be used to compute maps which put higher priority on the most meaningful feature dimensions. Note also that the mapping between source and target distributions used here is piecewise linear, and is therefore more regular.

**Conclusion and Future Work.** We have proposed in this paper a new class of transport plans and maps that are built using optimality constraints on a subspace, but defined over the whole space. We have presented two particular instances, MI and MK, with different properties, and derived closed formulations for Gaussian distributions. Future work includes exploring other applications of OT to machine learning relying on low-dimensional projections, from which subspace-optimal transport could be used to recover full-dimensional plans or maps.

## Footnotes

[1] Note that compared to [27], this is the reversed marginalization order, which is why the KR map here has *lower* triangular Jacobian.

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
