[Supplementary Material · Subspace_Detours_supplementary.pdf]

# 7 Supplementary Material

## 7.1 Proof of Proposition 2

*Proof.* The proof is a simpler, two-step variation of that of [5], which we refer to for additional details. For all $\varepsilon \geq 0$, let $\pi_\varepsilon$ be the optimal plan for $d^2_{\mathbf{P}_\varepsilon}$, and suppose there exists $\pi$ such that $\pi_\varepsilon \rightharpoonup \pi$ (which is possible up to subsequences). By definition of $\pi_\varepsilon$, we have that

$$\forall \varepsilon \geq 0, \int d^2_{\mathbf{P}_\varepsilon} \mathrm{d}\pi_\varepsilon \leq \int d^2_{\mathbf{P}_\varepsilon} \mathrm{d}\pi_{\mathrm{MK}}.$$

Since $d^2_{\mathbf{P}_\varepsilon}$ converges locally uniformly to $d^2_{\mathbf{V}_E} \stackrel{\mathrm{def}}{=} (x,y) \rightarrow (x-y)^\top \mathbf{V}_E \mathbf{V}_E^\top (x-y)$, we get $\int d^2_{\mathbf{V}_E} \mathrm{d}\pi \leq \int d^2_{\mathbf{V}_E} \mathrm{d}\pi_{\mathrm{MK}}$. But by definition of $\pi_{\mathrm{MK}}$, $(\pi_{\mathrm{MK}})_E \stackrel{\mathrm{def}}{=} (p_E, p_E)_\sharp \pi_{\mathrm{MK}}$ is the optimal transport plan on $E$, therefore the last inequality implies $\pi_E = (\pi_{\mathrm{MK}})_E$.

Next, notice that the $\pi_\varepsilon$'s all have the same marginals $\mu_E, \nu_E$ on $E$ and hence cannot perform better on $E$ than $\pi_{\mathrm{MK}}$. Therefore,

$$\int_{E \times E} d^2_{\mathbf{V}_E} \mathrm{d}(\pi_{\mathrm{MK}}) + \varepsilon \int d^2_{\mathbf{V}_{E^\perp}} \mathrm{d}\pi_\varepsilon \leq \int d^2_{\mathbf{P}_\varepsilon} \mathrm{d}\pi_\varepsilon$$

$$\leq \int d^2_{\mathbf{P}_\varepsilon} \mathrm{d}\pi_{\mathrm{MK}}$$

$$= \int_{E \times E} d^2_{\mathbf{V}_E} \mathrm{d}(\pi_{\mathrm{MK}})_E + \varepsilon \int d^2_{\mathbf{V}_{E^\perp}} \mathrm{d}\pi_{\mathrm{MK}}.$$

Hence, passing to the limit, $\int d^2_{\mathbf{V}_{E^\perp}} \mathrm{d}\pi \leq \int d^2_{\mathbf{V}_{E^\perp}} \mathrm{d}\pi_{\mathrm{MK}}$. Let us now disintegrate this inequality on $E \times E$ (using the equality $\pi_E = (\pi_{\mathrm{MK}})_E$):

$$\int\int_{E^\perp \times E^\perp} d^2_{\mathbf{V}_{E^\perp}} \mathrm{d}\pi_{(x_E, y_E)} \mathrm{d}(\pi_{\mathrm{MK}})_E \leq \int\int_{E^\perp \times E^\perp} d^2_{\mathbf{V}_{E^\perp}} \mathrm{d}(\pi_{\mathrm{MK}})_{(x_E, y_E)} \mathrm{d}(\pi_{\mathrm{MK}})_E.$$

Again, by definition, for $(x_E, y_E)$ in the support of $(\pi_{\mathrm{MK}})_E$, $(\pi_{\mathrm{MK}})_{(x_E, y_E)}$ is the optimal transportation plan between $\mu_{x_E}$ and $\nu_{y_E}$, and the previous inequality implies $\pi_{(x_E, y_E)} = (\pi_{\mathrm{MK}})_{(x_E, y_E)}$ for $(\pi_{\mathrm{MK}})_E$-a.e.$(x_E, y_E)$, and finally $\pi = \pi_{\mathrm{MK}}$. Finally, by the a.c. hypothesis, all transport plans $\pi_\varepsilon$ come from transport maps $T_\varepsilon$, which implies $T_\varepsilon \rightarrow T_{\mathrm{MK}}$ in $\mathrm{L}_2(\mu)$. ∎

## 7.2 Proof of Proposition 3

*Proof.* Let $\mathbf{X} \subset \mathbb{R}^d$ be a compact, $\mu, \nu \in \mathcal{P}(\mathbf{X})$ be two a.c. measures, $E$ a $k$-dimensional subspace which we identify w.l.o.g. with $\mathbb{R}^k$ and $\pi_{\mathrm{MI}} \in \mathcal{P}(\mathbb{R}^d \times \mathbb{R}^d)$ as in Definition 2. For $n \in \mathbb{N}$, let $\mu_n = \frac{1}{n}\sum_{i=1}^n \delta_{x_i}$, $\nu_n = \frac{1}{n}\sum_{i=1}^n \delta_{y_i}$ where the $x_i$ (resp. $y_i$) are i.i.d. samples from $\mu$ (resp. $\nu$). Let $t_n : \mathbb{R}^k \rightarrow \mathbb{R}^k$ be the Monge map from the projection on $E$ $(p_E)_\sharp \mu_n$ of $\mu_n$ to that of $\nu_n$, and $\pi_n \stackrel{\mathrm{def}}{=} (\mathrm{Id}, t_n)_\sharp [(p_E)_\sharp \mu_n]$.

Up to points having the same projections on $E$ (which under the a.c. assumption is a 0 probability event), $t_n$ can be extended to a transport between $\mu_n$ and $\nu_n$, whose transport plan we will denote $\gamma_n$.

Let $f \in C_b(\mathbf{X} \times \mathbf{X})$. Since we are on a compact, by density (given by the Stone-Weierstrass theorem) it is sufficient to consider functions of the form

$$f(x_1, ..., x_d; y_1, ..., y_d) = g(x_1, ..., x_k; y_1, ..., y_k)h(x_{k+1}, ..., x_d; y_{k+1}, ..., y_d).$$

We will use this along with the disintegrations of $\gamma_n$ on $E \times E$ (denoted $(\gamma_n)_{x_{1:k}, y_{1:k}}, (x_{1:k}, y_{1:k}) \in E \times E$) to prove convergence:

$$\int_{\mathbf{X} \times \mathbf{X}} f d\gamma_n = \int_{\mathbf{X} \times \mathbf{X}} g(x_{1:k}, y_{1:k})h(x_{k+1:d}, y_{k+1:d})d\gamma_n$$

$$= \int_{E \times E} g(x_{1:k}, y_{1:k})d\pi_n \int h(x_{k+1:d}, y_{k+1:d})d(\gamma_n)_{x_{1:k}, y_{1:k}}$$

$$= \int_{E \times E} g(x_{1:k}, y_{1:k})d\pi_n \int h(x_{k+1:d}, y_{k+1:d})d(\mu_n)_{x_{1:k}}d(\nu_n)_{t_n(x_{1:k})}.$$

Then, we use (i) the Arzela-Ascoli theorem to get uniform convergence of $t_n$ to $T_E$ to get $d(\nu_n)_{t_n(x_{1:k})} \rightharpoonup d(\nu)_{T_E(x_{1:k})}$ and (ii) the convergence $\pi_n \rightharpoonup (p_E, p_E)_\sharp(\pi_{\mathrm{MI}})$ to get

$$\int_{E \times E} g(x_{1:k}, y_{1:k}) d\pi_n \int h(x_{k+1:d}, y_{k+1:d}) d(\mu_n)_{x_{1:k}} d(\nu_n)_{t_n(x_{1:k})}$$

$$\to \int_{E \times E} g(x_{1:k}, y_{1:k}) d(p_E, p_E)_\sharp(\pi_{\mathrm{MI}}) \int h(x_{k+1:d}, y_{k+1:d}) d(\mu)_{x_{1:k}} d(\nu)_{T_E(x_{1:k})}$$

$$= \int_{\mathbf{X} \times \mathbf{X}} f d\pi_{\mathrm{MI}},$$

which concludes the proof in the compact case. $\blacksquare$

## 7.3 Proof of Proposition 4

*Proof.* Let $\mathbf{T}_E : \mathbf{A}_E^{-\frac{1}{2}} (\mathbf{A}_E^{\frac{1}{2}} \mathbf{B}_E \mathbf{A}_E^{\frac{1}{2}})^{\frac{1}{2}} \mathbf{A}_E^{-\frac{1}{2}}$ be the Monge map from $\mu_E \stackrel{\mathrm{def}}{=} (p_E)_\sharp \mu$ and $\nu_E \stackrel{\mathrm{def}}{=} (p_E)_\sharp \nu$. Let

$$V = \begin{pmatrix} | & & | & | & & | \\ v_1 & \cdots & v_k & v_{k+1} & \cdots & v_d \\ | & & | & | & & | \end{pmatrix} = (\mathbf{V}_E \quad \mathbf{V}_{E^\perp}) \in \mathbb{R}^{d \times d},$$

where $(v_1 \dots v_k)$ is an orthonormal basis of $E$ and $(v_{k+1} \dots v_d)$ an orthonormal basis of $E^\perp$. Let us denote $X_E \stackrel{\mathrm{def}}{=} p_E(X) \in \mathbb{R}^k$ and *mutatis mutandis* for $Y, E^\perp$. Denote $\mathbf{A}_E = p_E \mathbf{A} p_E^\top$, $\mathbf{A}_{E^\perp} = p_{E^\perp} \mathbf{A} p_{E^\perp}^\top$, $\mathbf{A}_{EE^\perp} = p_E \mathbf{A} p_{E^\perp}^\top$. With these notations, we decompose the derivation of $\mathbb{E}[XY^\top]$ along $E$ and $E^\perp$:

$$\mathbb{E}[XY^\top] = \mathbb{E}[\mathbf{V}_E X_E (\mathbf{V}_E Y_E)^\top] + \mathbb{E}[\mathbf{V}_{E^\perp} X_{E^\perp} (\mathbf{V}_{E^\perp} Y_{E^\perp})^\top]$$
$$+ \mathbb{E}[\mathbf{V}_{E^\perp} X_{E^\perp} (\mathbf{V}_E Y_E)^\top]$$
$$+ \mathbb{E}[\mathbf{V}_E X_E (\mathbf{V}_{E^\perp} Y_{E^\perp})^\top].$$

We can condition all four terms on $X_E$, and use point independence given coordinates on $E$ which implies $(Y_E | X_E) = X_E$. The constraint $Y_E = \mathbf{T}_E X_E$ allows us to derive $\mathbb{E}[Y_{E^\perp} | X_E]$: indeed, it holds that

$$\begin{pmatrix} Y_E \\ Y_{E^\perp} \end{pmatrix} \sim \mathcal{N}\left(0_d, \begin{pmatrix} \mathbf{B}_E & \mathbf{B}_{EE^\perp} \\ \mathbf{B}_{EE^\perp}^\top & \mathbf{B}_{E^\perp} \end{pmatrix}\right),$$

which, using standard Gaussian conditioning properties, implies that

$$\mathbb{E}[Y_{E^\perp} | Y_E = \mathbf{T}_E X_E] = \mathbf{B}_{EE^\perp}^\top \mathbf{B}_E^{-1} \mathbf{T}_E X_E,$$

and therefore

$$\mathbb{E}[Y_{E^\perp} | \mathbf{P}_E(Y) = \mathbf{T}_E X_E] = V_{E^\perp} \mathbf{B}_{EE^\perp}^\top \mathbf{B}_E^{-1} \mathbf{V}_E^\top \mathbf{T}_E X_E.$$

Likewise,

$$\mathbb{E}[X_{E^\perp} | \mathbf{P}_E(X)] = \mathbf{V}_{E^\perp} \mathbf{A}_{EE^\perp}^\top \mathbf{A}_E^{-1} \mathbf{V}_E^\top X_E.$$

We now have all the ingredients necessary to the derivation of the four terms of $\mathbb{E}[XY^\top]$:

$$\mathbb{E}[\mathbf{V}_E X_E Y_E^\top \mathbf{V}_E^\top] = \mathbf{V}_E \mathbb{E}_{X_E} \left[ \mathbb{E}\left[ X_E Y_E^\top | X_E \right] \right] \mathbf{V}_E^\top$$
$$= \mathbf{V}_E \mathbb{E}_{X_E} \left[ X_E \mathbb{E}\left[ Y_E^\top | X_E \right] \right] \mathbf{V}_E^\top$$
$$= \mathbf{V}_E \mathbb{E}_{X_E} \left[ X_E X_E^\top \mathbf{T}_E^\top \right] \mathbf{V}_E^\top$$
$$= \mathbf{V}_E \mathbb{E}_{X_E} \left[ X_E X_E^\top \right] \mathbf{T}_E^\top \mathbf{V}_E^\top$$
$$= \mathbf{V}_E \mathbf{A}_E \mathbf{T}_E \mathbf{V}_E^\top$$

$$\mathbb{E}[\mathbf{V}_E X_E Y_{E^\perp}^\top \mathbf{V}_{E^\perp}^\top] = \mathbf{V}_E \mathbb{E}_{X_E} \left[ \mathbb{E}[X_E Y_{E^\perp}^\top | X_E] \mathbf{V}_{E^\perp}^\top \right.$$
$$= \mathbf{V}_E \mathbb{E}_{X_E} \left[ X_E \mathbb{E}\left[ Y_{E^\perp}^\top | X_E = \mathbf{T}_E X_E \right] \right] \mathbf{V}_{E^\perp}^\top$$
$$= \mathbf{V}_E \mathbb{E}_{X_E} \left[ X_E \left( V_{E^\perp} \mathbf{B}_{EE^\perp}^\top \mathbf{B}_E^{-1} \mathbf{V}_E^\top \mathbf{T}_E X_E) \right)^\top \right] \mathbf{V}_{E^\perp}^\top$$
$$= \mathbf{V}_E \mathbb{E}_{X_E} \left[ X_E X_E^\top \right] \mathbf{T}_E^\top \mathbf{V}_E \mathbf{B}_{V_E}^{-\top} \mathbf{B}_{V_{EE^\perp}} \mathbf{V}_{E^\perp}^\top$$
$$= \mathbf{V}_E \mathbf{A}_E \mathbf{T}_E \mathbf{V}_E \mathbf{B}_E^{-1} \mathbf{B}_{V_{EE^\perp}} \mathbf{V}_{E^\perp}^\top$$
$$= \mathbf{V}_E \mathbf{A}_E \mathbf{T}_E \mathbf{V}_E \mathbf{B}_E^{-1} \mathbf{V}_E^\top \mathbf{B}_{EE^\perp} \mathbf{V}_{E^\perp}^\top$$

$$\mathbb{E}[\mathbf{V}_{E^\perp} X_{E^\perp} Y_E^\top \mathbf{V}_E^\top] = \mathbf{V}_{E^\perp} \mathbb{E}_{X_E} \left[ \mathbb{E}[X_{E^\perp} Y_E^\top | X_E] \mathbf{V}_E^\top \right.$$
$$= \mathbf{V}_{E^\perp} \mathbb{E}_{X_E} \left[ \mathbb{E}\left[ X_{E^\perp} | X_E \right] X_E^\top \mathbf{T}_E^\top \right] \mathbf{V}_E^\top$$
$$= \mathbf{V}_{E^\perp} \mathbb{E}_{X_E} \left[ \mathbf{A}_{EE^\perp}^\top \mathbf{A}_E^{-1} X_E X_E^\top \mathbf{T}_E^\top \right] \mathbf{V}_E^\top$$
$$= \mathbf{V}_{E^\perp} \mathbf{V}_{E^\perp} \mathbf{A}_{EE^\perp}^\top \mathbf{A}_E^{-1} \mathbf{V}_E^\top \mathbf{A} \mathbf{T}_E \mathbf{V}_E^\top$$
$$= \mathbf{V}_{E^\perp} \mathbf{V}_{E^\perp} \mathbf{A}_{EE^\perp}^\top \mathbf{T}_E \mathbf{V}_E^\top$$
$$= \mathbf{V}_{E^\perp} \mathbf{A}_{EE^\perp}^\top \mathbf{T}_E \mathbf{V}_E^\top$$

$$\mathbb{E}[\mathbf{V}_{E^\perp} X_{E^\perp} Y_{E^\perp}^\top \mathbf{V}_{E^\perp}^\top] = V_{E^\perp E} \mathbb{E}_{X_E} \left[ \mathbb{E}[X_{E^\perp} | X_E] \mathbb{E}\left[ Y_{E^\perp}^\top | X_E \right] \mathbf{V}_{E^\perp}^\top \right.$$
$$= \mathbf{V}_{E^\perp} \mathbb{E}_{X_E} \left[ \mathbf{V}_{E^\perp} \mathbf{A}_{EE^\perp}^\top \mathbf{A}_E^{-1} \mathbf{V}_E^\top X_E X_E^\top \mathbf{T}_E^\top \mathbf{V}_E \mathbf{B}_{V_E}^{-\top} \mathbf{B}_{EE^\perp} \right] \mathbf{V}_{E^\perp}^\top$$
$$= \mathbf{V}_{E^\perp} \mathbf{A}_{EE^\perp}^\top \mathbf{A}_E^{-1} \mathbf{V}_E^\top \mathbf{A}_E \mathbf{T}_E \mathbf{V}_E \mathbf{B}_E^{-1} \mathbf{B}_{EE^\perp} \mathbf{V}_{E^\perp}^\top$$
$$= \mathbf{V}_{E^\perp} \mathbf{A}_{EE^\perp}^\top \mathbf{T}_E \mathbf{B}_E^{-1} \mathbf{B}_{EE^\perp} \mathbf{V}_{E^\perp}^\top$$
$$= V_{E^\perp} \mathbf{A}_{EE^\perp}^\top \mathbf{T}_E \mathbf{V}_E \mathbf{B}_{V_E}^{-1} \mathbf{V}_E^\top \mathbf{B}_{EE^\perp},$$

Let $\gamma \stackrel{\text{def}}{=} \mathcal{N}(0_{2d}, \Sigma_{\pi_E})$. $\gamma$, is well defined, since $\Sigma_{\pi_E}$ is the covariance matrix of $\pi_E$ and is thus PSD. From then, $\gamma$ clearly has marginals $\mathcal{N}(0_d, \mathbf{A})$ and $\mathcal{N}(0_d, \mathbf{B})$, and is such that $(p_E, p_E)_\sharp \gamma$ is a centered Gaussian distribution with covariance matrix

$$\begin{pmatrix} p_E & 0_{d \times d} \\ 0_{d \times d} & p_E \end{pmatrix} \begin{pmatrix} \mathbf{A} & \mathbb{E}_\pi[XY^\top] \\ \mathbb{E}_\pi[YX^\top] & \mathbf{B} \end{pmatrix} \begin{pmatrix} p_E & 0_{d \times d} \\ 0_{d \times d} & p_E \end{pmatrix} = \begin{pmatrix} \mathbf{A}_E & \mathbf{A}_E \mathbf{T}_E \\ \mathbf{T}_E \mathbf{A}_E & \mathbf{B}_E \end{pmatrix},$$

where we use that $p_E p_E = p_E$ and $p_E p_{E^\perp} = 0$. From the $k = d$ case, we recognise the covariance matrix of the optimal transport between centered Gaussians with covariance matrices $\mathbf{A}_E$ and $\mathbf{B}_E$, which proves that the marginal of $\gamma$ over $E \times E$ is the optimal transport between $\mu_E$ and $\nu_E$.

To complete the proof, there remains to show that the disintegration of $\gamma$ on $E \times E$ is the product law. Denote

$$\mathbf{C} \stackrel{\text{def}}{=} \mathbb{E}[XY^\top]$$
$$= \mathbf{V}_E \mathbf{A}_E \mathbf{T}_E \left( \mathbf{V}_E^\top + (\mathbf{B}_E)^{-1} \mathbf{V}_E^\top \mathbf{B}_{EE^\perp} \right) + \mathbf{V}_{E^\perp} \mathbf{A}_{E^\perp E} \mathbf{T}_{V_E} \left( \mathbf{V}_E^\top + (\mathbf{B}_{V_E})^{-1} \mathbf{V}_E^\top \mathbf{B}_{EE^\perp} \right)$$
$$= \left( \mathbf{V}_E \mathbf{A}_E + \mathbf{V}_{E^\perp} \mathbf{A}_{E^\perp E} \right) \mathbf{T}_E \left( \mathbf{V}_E^\top + (\mathbf{B}_E)^{-1} \mathbf{B}_{EE^\perp} \mathbf{V}_{E^\perp}^\top \right),$$

and let $\Sigma_{\pi_{\mathrm{MI}}} = \begin{pmatrix} \mathbf{A} & \mathbb{E}[XY^\top] \\ \mathbb{E}[YX^\top] & \mathbf{B} \end{pmatrix}$ as in Prop. 4. It holds that

$$\mathbf{C}_E \overset{\mathrm{def}}{=} \mathbf{V}_E^\top \mathbf{C} \mathbf{V}_E = \mathbf{A}_E \mathbf{T}_E$$

$$\mathbf{C}_{E^\perp} \overset{\mathrm{def}}{=} \mathbf{V}_{E^\perp}^\top \mathbf{C} \mathbf{V}_E = \mathbf{A}_{E^\perp E} \mathbf{T}_E (\mathbf{B}_E)^{-1} \mathbf{B}_{EE^\perp}$$

$$\mathbf{C}_{EE^\perp} \overset{\mathrm{def}}{=} \mathbf{V}_E^\top \mathbf{C} \mathbf{V}_{E^\perp} = \mathbf{A}_E \mathbf{T}_E (\mathbf{B}_E)^{-1} \mathbf{B}_{EE^\perp}$$

$$\mathbf{C}_{E^\perp E} \overset{\mathrm{def}}{=} \mathbf{V}_{E^\perp}^\top \mathbf{C} \mathbf{V}_E = \mathbf{A}_{E^\perp E} \mathbf{T}_E.$$

Therefore, if $(X, Y) \sim \gamma$, then

$$\mathrm{Cov} \begin{pmatrix} X_{E^\perp} \\ Y_{E^\perp} \\ X_E \\ Y_E \end{pmatrix} = \begin{pmatrix} \mathbf{A}_{E^\perp} & \mathbf{C}_{E^\perp} & \mathbf{A}_{E^\perp E} & \mathbf{C}_{E^\perp E} \\ \mathbf{C}_{E^\perp} & \mathbf{B}_{E^\perp} & \mathbf{C}_{EE^\perp}^\top & \mathbf{B}_{E^\perp E} \\ \mathbf{A}_{EE^\perp} & \mathbf{C}_{EE^\perp} & \mathbf{A}_E & \mathbf{C}_E \\ \mathbf{C}_{E^\perp E}^\top & \mathbf{B}_{EE^\perp} & \mathbf{C}_E & \mathbf{B}_E \end{pmatrix},$$

and therefore

$$\mathrm{Cov} \begin{pmatrix} X_{E^\perp} & | X_E \\ Y_{E^\perp} & | Y_E \end{pmatrix} = \begin{pmatrix} \mathbf{A}_{E^\perp} & \mathbf{C}_{E^\perp} \\ \mathbf{C}_{E^\perp} & \mathbf{B}_{E^\perp} \end{pmatrix} - \begin{pmatrix} \mathbf{A}_{E^\perp E} & \mathbf{C}_{E^\perp E} \\ \mathbf{C}_{EE^\perp}^\top & \mathbf{B}_{E^\perp E} \end{pmatrix} \begin{pmatrix} \mathbf{A}_E & \mathbf{C}_E \\ \mathbf{C}_E & \mathbf{B}_E \end{pmatrix}^\dagger \begin{pmatrix} \mathbf{A}_{EE^\perp} & \mathbf{C}_{EE^\perp} \\ \mathbf{C}_{E^\perp E}^\top & \mathbf{B}_{EE^\perp} \end{pmatrix},$$

where $\mathbf{M}^\dagger$ denotes the Moore-Penrose pseudo-inverse of $\mathbf{M}$. In the present case, one can check that

$$\begin{pmatrix} \mathbf{A}_E & \mathbf{C}_E \\ \mathbf{C}_E & \mathbf{B}_E \end{pmatrix}^\dagger = \frac{1}{4} \begin{pmatrix} \mathbf{A}_E^{-1} & \mathbf{A}_E^{-1} \mathbf{T}_E^{-1} \\ \mathbf{T}_E^{-1} \mathbf{A}_E^{-1} & \mathbf{B}_E^{-1} \end{pmatrix},$$

which gives, after simplification

$$\begin{pmatrix} \mathbf{A}_{E^\perp E} & \mathbf{C}_{E^\perp E} \\ \mathbf{C}_{EE^\perp}^\top & \mathbf{B}_{E^\perp E} \end{pmatrix} \begin{pmatrix} \mathbf{A}_E & \mathbf{C}_E \\ \mathbf{C}_E & \mathbf{B}_E \end{pmatrix}^\dagger \begin{pmatrix} \mathbf{A}_{EE^\perp} & \mathbf{C}_{EE^\perp} \\ \mathbf{C}_{E^\perp E}^\top & \mathbf{B}_{EE^\perp} \end{pmatrix} = \begin{pmatrix} \mathbf{A}_{E^\perp E} \mathbf{A}_E^{-1} \mathbf{A}_{EE^\perp} & \mathbf{C}_{E^\perp} \\ \mathbf{C}_{E^\perp} & \mathbf{B}_{E^\perp E} \mathbf{B}_E^{-1} \mathbf{B}_{EE^\perp} \end{pmatrix},$$

and thus

$$\mathrm{Cov} \begin{pmatrix} X_{E^\perp} & | X_E \\ Y_{E^\perp} & | Y_E \end{pmatrix} = \begin{pmatrix} \mathbf{A}_{E^\perp} - \mathbf{A}_{E^\perp E} (\mathbf{A}_E)^{-1} \mathbf{A}_{EE^\perp} & 0_d \\ 0_d & \mathbf{B}_{E^\perp} - \mathbf{B}_{E^\perp E} (\mathbf{B}_E)^{-1} \mathbf{B}_{EE^\perp} \end{pmatrix}$$

$$= \begin{pmatrix} \mathrm{Cov}(X_{E^\perp} | X_E) & 0_d \\ 0_d & \mathrm{Cov}(Y_{E^\perp} | Y_E) \end{pmatrix},$$

that is, the conditional laws of $X_{E^\perp}$ given $X_E$ and $Y_{E^\perp}$ given $Y_E$ are independent under $\gamma$.

■