[Reviews · NeurIPS 2019]

Reviewer 1



Post-response comments (from discussion): "I feel the response did a good job of answering points of confusion, and also added an interesting example application of color transfer. This last example/experiment is heartening, because they finally use one of their maps (MK) in an application instead of just using the distances. I would be quite interested to see a more comprehensive exploration of this, as well as further applications of the MI and MK maps (perhaps in domain adaptation or Waddington-OT, which they mention elsewhere?). It's also important to note that they included a new algorithm for subspace selection which performs projected gradient descent on the basis vectors of the subspace, which outperforms their old method in the synthetic cases. This is a nice discovery, but I think will necessitate more than a minor structural change to the paper. One would expect a more complete presentation of the algorithm, including discussion of convergence (to local optima at least?) and runtime. It would also be nice to find a non-synthetic use case for this subspace selection, if possible. In light of their response, I feel that this paper is on the right track, but could use another iteration to better argue for applicability of their maps, and to update their algorithm for subspace selection." ===================================================== This paper presents an interesting fundamental concept that naturally arises when one considers subspace-restricted transport problems. I was not able to check line-by-line all the mathematics, but I feel it performs a careful basic exploration of these concepts. However, as hinted at above, the main weakness of the paper is its applied facet. It presents a subspace selection algorithm without any real analysis or experimental exploration. The applications presented seem fairly esoteric and don't seem to really demonstrate the utility of the full transports obtained (they explore mainly the distances obtained).

Reviewer 2



Comments: 1: A minor critique: is E introduced before line 44? It is clear from later reading and after studying the context that E denotes a particular subspace, but the notation section follows after this. 2: Line 337, is this counter-example demonstrating something other than the analogy that performing PCA with a (random) subset other than the first k components leads to sub-optimal outcomes? 3: Line 333: what does ‘underestimated’ mean in this context? Moreover, in this setting, does the number of samples used to estimate the covariance matrices (n) equal the dimensionality (p) (i.e., does p = n)? If so, would one expect the covariance matrices estimated to be reliable or of any decent quality? For example, in the low-rank-plus-noise setting, one can end up with estimated principal components that are asymptotically orthogonal to the true components when p/n does not go to 0. See (for example) Johnstone and Lu, 2009 4: Continuing from (3), more generally, what is the value of d_2 in the simulations shown in Figure 4? Have you investigated the role of d_2 in this context?

Reviewer 3



The lifting technique involves the disintegration of measure. The whole construction of MI and MK is quite simple, so we can say that it is a technically simple theoretical paper. Quality of that approach is strongly dependant on how we choose E, so one chapter is dedicated to this issue. I could not find experimental verification of that chapter's suggestions, especially of Algorithm 1? Experiments with synthetic data seems informative, but semantic mediation etc are not convincing.

[Author Response · NeurIPS 2019]

We thank reviewers for carefully reading our paper. We answer their questions

**Algorithm 1** MK Projected GD

**Input:** $\mathbf{A}, \mathbf{B} \in \mathrm{PSD}, k \in [\![1, d]\!], \eta$
   $\mathbf{V} \leftarrow \mathrm{Polar}(\mathbf{AB})$
   **while** not converged **do**
      $\mathcal{L} \leftarrow \mathrm{MK}(\mathbf{V}^\top \mathbf{A} \mathbf{V}, \mathbf{V}^\top \mathbf{B} \mathbf{V}; k)$
      $\mathbf{V} \leftarrow \mathbf{V} - \eta \nabla_{\mathbf{V}} \mathcal{L}$
      $\mathbf{V} \leftarrow \mathrm{Polar}(\mathbf{V})$
   **end while**
**Output:** $E = \mathrm{Span}\{\mathbf{v}_1, .., \mathbf{v}_k\}$

below, but provide first two updates that are directly related to their remarks.
▶ *Subspace Selection:* Alg. 1 from the paper was motivated by Prop. 6 (l.
263). After benchmarking it carefully, we now believe it is not competitive
with a projected gradient descent (PGD) on the basis vectors $\mathbf{V}$ of $E$ (see
right). The projection of $\mathbf{V}$ onto the set of unitary matrices is the unitary
matrix in the polar decomposition of $\mathbf{V}$. The complexity per iteration is that
of computing MK and the polar decomposition. We initialize $V = \mathrm{Polar}(\mathbf{AB})$
because this is the optimal solution when $\mathbf{A}, \mathbf{B}$ are co-diagonalizable. We
tested this new algo. in the synthetic noisy setting (p.7), Fig.1 below. The
PGD improves on the fixed direction (canonical basis) approach when $k < 4$, and remains competitive when $k \geq 4$.
▶ *Map visualization using color transfer:* All reviewers have pointed out that experiments in the paper did not illustrate
the lifted transport maps/plans, but focused instead on distances. We experimented MK maps on color transfer, an
illustrative task to visualize maps' properties. In the MK setting, we project images on the 1D space of grayscale images,
relying on sorting-based algos for 1D-OT, before solving small 2D-OT problems on the corresponding *disintegrations*.
We compare runtimes and visual results with vanilla OT and sliced OT below. MK results are visually very similar to
full OT, with a $\sim \times 50$ speedup that is comparable to sliced OT. We will provide other illustrations.

Figure 1: Synthetic data experiment (p.7): canonical directions vs PGD

Reviewer #1: ▶ *algo in terms of*
*optimality, convergence, runtime,*
*etc.* The runtime involves a com-
plexity per iteration equal to com-
puting the polar decomp. and MK
distance + gradient. Because the
problem is non-convex we will
stick to empirical evaluations and
improve the presentation (p.7), as
in Fig. 1 (right). ▶ *applications do not seem to be terribly important [...] more popular ones.* Agreed. Color transfer
was added as an illustrative example. We are now looking into applications to domain adaptation and biological datasets
(Waddington-OT). ▶ *experiments section [...] a little confusing.* We will add more context. The main purpose of the
FID exp. (p.8) is to use data widely handled as samples from Gaussians. We show that even with a relatively small
number of samples to estimate the covariance matrices, MK on the principal components has a stable behavior.

Reviewer #2: ▶ *1:* $E$ is indeed introduced later, l.62. We will fix this.
▶ *2: PCA with a (random) subset.* This counter-example is to show
that the stability of MK is dependent on the chosen subspace. Permut-
ing the principal directions is an adversarial setting used to showcase
this. ▶ *3: what does 'underestimated' mean [...] covariance matrices*
*estimated [...] decent quality?* In the setting of FID (p.8), $p = 2048$
and we used $n = 2050$. Fig. 2 (right) shows the covergence of
sample to full (on all 200K data points) covariance matrices in Bures
and L2 distance (averaged over 20 sample matrices). At $n = 2050$
the sample covariance matrices are close to having converged but
not quite. However, the MK distance on the principal components
is robust to the small amount of noise thus induced. We are glad to

Figure 2: Mean distances from sample matrices to full covariance matrix (FID setting, p.8)

include this point in the discussion. ▶ *4: [...] value of $d_2$ [...] role of $d_2$ in this context?* As per the caption in the paper
(Fig.4, p.7) $d_1 = 4$, top row is $d_2 = 8$ and bottom row $d_2 = 16$. We will make this more explicit. As $d_2$ increases, the
MK distance for $d_1 \leq k \leq d_2$ increases as more noise is fitted by the transport map on the projection subspace.
Reviewer #3: ▶ *experimental verification of that chapter's suggestions, especially of Algo. 1?* Semantic mediation
(p.8) is an example of using MK with prescribed directions (l.242-249), and FID experiments (p.8) of using principal
components. We have added a verification of the new PGD algo in the experiment on noisy data (Fig. 1). ▶ *Experiments*
*with synthetic data seems informative, but semantic mediation etc are not convincing.* We added more semantic
mediation examples. We are considering domain adaptation and biological datasets. ▶ *Experiments on real data, and*
*some more attention to selection of subspace $E$ (experimentally).* Agreed. The PGD approach is a first step in that
direction (Fig. 1). We will also try it first in color transfer, domain adaptation and in biology (Waddington-OT).

[Meta-Review · NeurIPS 2019]

This work generated significant discussion but in the end we decided to accept it, modulo some revision on your part. As discussed in the rebuttal phase, be sure to do the following: ** Change the algorithm for subspace selection and replace the existing results for the current one. Comment on convergence guarantees (if any) and runtime of the algorithm. ** Include the color transfer application experiments from the rebuttal. As pointed out in the reviews, the main weakness of this work is that it does not include applications of the MK/MI **maps** computed using the algorithm (as opposed to applications of the objective value). If there are other simple applications you can include in the final revision, this would strengthen the work considerably. And of course in your revision be sure to edit responsively to other comments/typos identified in the reviews.